



# Global impact of monocyclic aromatics on tropospheric composition

David Cabrera-Perez[1], Domenico Taraborrelli[2], Jos Lelieveld[1],
Thorsten Hoffmann[3], and Andrea Pozzer[1]

[1]Atmospheric Chemistry Department, Max-Planck Institute of Chemistry, Hahn-Meitner-Weg 1,
D-55128 Mainz, Germany
[2]Institute of Energy and Climate Research (IEK-8), Forschungszentrum Jülich GmbH, Jülich,
Germany
[3]Institute of Inorganic and Analytical Chemistry, University of Mainz, Duesbergweg 10–14, 55128
Mainz, Germany

*Correspondence to:* D. Cabrera (d.cabrera@mpic.de)

**Abstract.** Aromatic compounds are reactive species influencing ozone formation, OH concentrations and organic aerosol formation. An assessment of their impacts on the gas-phase composition at a global scale has been performed using a general circulation atmospheric-chemistry model.

Globally, we found a small annual average net decrease (less than 3%) in global OH, ozone, and $NO_x$ mixing ratios when aromatic compounds are included in the chemical mechanism. This inclusion of aromatics also results in CO mixing ratio increases, which cause a general decrease in OH concentrations. The largest changes are found in glyoxal and $NO_3$, with increases in the atmospheric burden of 10% and 6%, respectively.

Regionally, significant differences were found particularly in high $NO_x$ regime areas, with an
increase of up to 4% in $O_3$ mixing ratios and 8% in OH concentrations. $NO_3$ increased by more than 30% in several regions of the northern hemisphere, and glyoxal increased up to 40% in Europe and Asia. Large increases in formaldehyde were found in urban areas.

Although the relative impact of aromatics at the global scale is limited, at a regional level they are important in atmospheric chemistry.

## 1 Introduction

Volatile organic compounds (VOCs) comprises a large variety of compounds whose oxidation influences the tropospheric chemistry at local, regional, and global scales. The VOC oxidation affects the formation of key atmospheric species, for example OH, which controls the tropospheric oxidative capacity (Levy, 1971; Koppmann, 2008); and ozone, a major atmospheric pollutant with the capa-
bility of modifying the climate forcing and toxic for terrestrial life (Lelieveld and Dentener, 2000; Finlayson-Pitts and Pitts, 1997; Forster et al., 2007; Lelieveld et al., 2015).



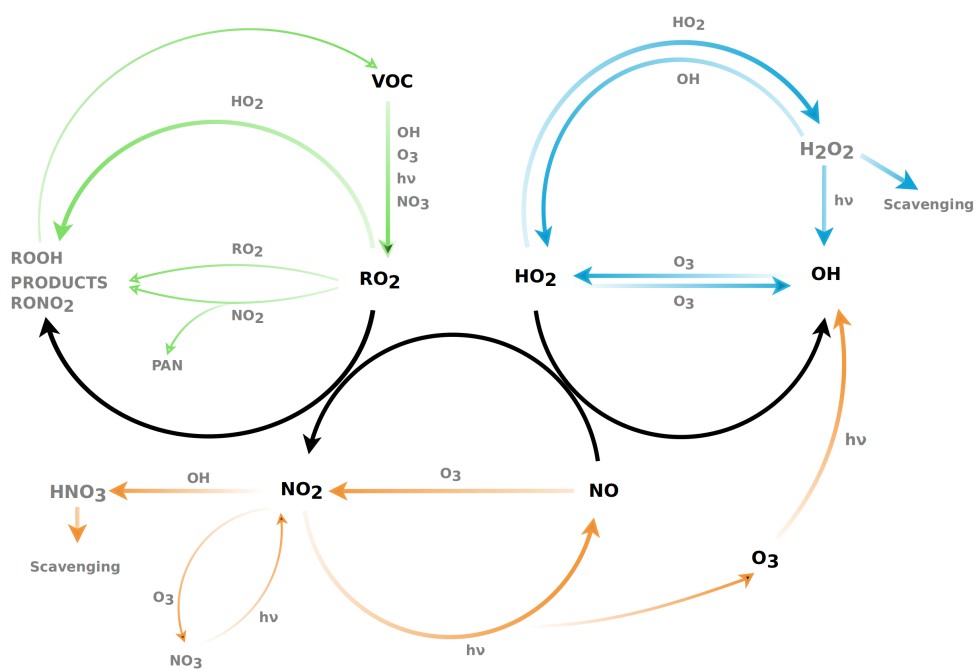

**Figure 1.** Schematic of the main recycling process of $NO_x$ (orange), $HO_x$ (blue), and VOC degradation (green). Black lines denote the interaction between the different cycles.

In general, ozone formation is controlled by the presence of VOCs and $NO_x$ ($NO + NO_2$). VOCs are oxidized by $O_3$, OH, $NO_3$ or photolyzed yielding peroxy radicals ($RO_2$ in reaction R1).

$$VOC + OH \rightarrow RO_2 \tag{R1}$$

The reaction of NO either with the hydroperoxyl ($HO_2$) or the peroxy radical ($RO_2$) produces $NO_2$.

$$HO_2 + NO \rightarrow NO_2 + OH \tag{R2}$$

$$RO_2 + NO \rightarrow NO_2 + RO \tag{R3}$$

The $NO_x$ cycle is completed by $NO_2$ reacting with OH to form nitric acid ($HNO_3$), which can be scavenged or be subject to slow photodissociation, making $HNO_3$ a relatively long-lived compound and the end of the NO chain (Finlayson-Pitts and Pitts Jr, 1986). The photolysis of $NO_2$ leads to $O_3$ formation and NO recycling.

$$NO_2 + h\nu + (O_2) \rightarrow O_3 + NO \tag{R4}$$






$$O_3 + h\nu \rightarrow O(^1D) + O_2 \qquad (R5)$$

$$O(^1D) + H_2O \rightarrow 2OH \qquad (R6)$$

The reactions R4-R6 form a major path of ozone formation in the troposphere and the efficiency of
this catalytic process is dependent on the VOC and $NO_x$ concentrations and their ratio (Kleinman,
1994).

Simultaneously to the $NO_x$ cycle, in the $HO_x$ ($HO_2 + OH$) cycle OH radicals react with $O_3$ to
form $HO_2$. The reaction of $O_3$ with $HO_2$ produces OH.

$$OH + O_3 \rightarrow HO_2 + (O_2) \qquad (R7)$$


$$HO_2 + O_3 \rightarrow OH + (2O_2) \qquad (R8)$$

Self-reaction of $HO_2$ produces $H_2O_2$ (reaction R9), which under photolysis recycles OH (reaction
R10). However, the efficiency of this recycling process is limited because $H_2O_2$ is rapidly scavenged.

$$H_2O_2 + OH \rightarrow HO_2 + (H_2O) \qquad (R9)$$


$$H_2O_2 + h\nu \rightarrow 2OH \qquad (R10)$$

As $HO_x$ and $NO_x$ cycles and the VOC degradation are interconnected, changes in VOC concentra-
tions directly affect the balance and feedback between the cycles. A general scheme describing the
VOC, $NO_x$ and $HO_x$ cycles and how they are interconnected is presented in Fig. 1.

In the low $NO_x$ regime, ozone production is limited by the reactive nitrogen, independent of the
VOC concentrations. Under this condition, net ozone loss takes place; this regime can be consid-
ered the default regime for low $NO_x$ environments. In contrast, the high $NO_x$ regime is limited
by the VOC concentration. This regime describes urban environments, which results in net ozone
formation.

Among VOCs, aromatics are a group of major relevance in urbanized and industrialized environ-
ments. In general, aromatic compounds are present in continental areas, and usually high mixing
ratios are observed in urban regions (Barletta et al., 2005). Because of their high reactivities, they
have relative short lifetimes—ranging from hours to a few days. The emissions are primarily an-
thropogenic, being mainly fossil fuel combustion and leakage, and solvent as the most important


sources (Koppmann, 2008; Sack et al., 1992). Emissions from biomass burning play a secondary role, but can be important at the regional scale (Cabrera-Perez et al., 2016). Biogenic emissions are only relevant for toluene, although recent studies suggest that aromatics from biogenic sources may rival those from fossil fuel (Misztal et al., 2015).

The volatility of aromatic compounds is high, although upon oxidation they present a large source 70  of secondary organic aerosol (SOA) (Odum et al., 1996; Ziemann and Atkinson, 2012). In urban and semi-urban areas aromatic compounds are a group of major importance for tropospheric chemistry, as they are responsible for a considerable fraction of the ozone and SOA formation (Ng et al., 2007; Lee et al., 2002; Ran et al., 2009). Furthermore, they are toxic compounds (WMO, 2000) and can affect directly and indirectly human health.

The gas phase chemistry of aromatics is relatively well known, as it is supported by laboratory studies and field campaigns (Atkinson et al., 1989; Zou et al., 2003; Jenkin et al., 2003; Baker et al., 2008). Aromatic compound concentrations are controlled by the hydroxyl radical (OH), but they can also react with nitrate ($NO_3$) and ozone ($O_3$) radicals, for instance styrene, of which the main sink is reaction with nitrate radicals (Atkinson, 2000). The complete oxidation process from the aromatic 80  compounds to its full degradation to $CO_2$ is rather complex, and during this process ozone, nitrates, $NO_x$ and $HO_x$ are formed and consumed.

Due to the highly complex chemical degradation of aromatics, numerical models are essential to quantify the impact of these compounds in atmospheric chemistry. However, at the global scale studies concerning the gas-phase atmospheric implication of aromatic compounds are lacking. The 85  main objective of this study is to disentangle how tropospheric $HO_x$, $NO_x$, $O_3$, and several VOC concentrations are affected globally by the oxidation of aromatic compounds. For this task we use a global atmospheric circulation model and a series of sensitivity simulations.

A detailed description of the model, set-up and scenarios used for this study are described in section 2. Section 3 presents model results for ozone, OH and VOCs comparing two scenarios and 90  estimating the influence of aromatics. Different sources of error are discussed in sect. 4 followed by the conclusions (sect. 5).

## 2   Model set-up

The model employed for this study is the ECHAM5 MESSy Atmospheric Chemistry (EMAC) model. The Modular Earth Submodel System (MESSy) is a modularized system which allows 95  the representation of atmospheric photo-chemistry, deposition, transport, radiation and cloud atmospheric processes (Jöckel et al., 2010). MESSy (version 2.50) is coupled to the general circulation model ECHAM5 (version 5.3.02) (Roeckner et al., 2006). EMAC is an atmospheric chemistry general circulation model and it has been extensively evaluated (Pozzer et al., 2007, 2010).


**Table 1.** List of aromatic compounds included in this study and the respective annual emissions. This emissions are the same as in (Cabrera-Perez et al., 2016) but for higher aromatics.

| Species | Emissions (Tg/yr) |
|---|---|
| Benzene | 8.5 |
| Toluene | 8.7 |
| Xylenes | 6.6 |
| Ethylbenzene | 0.9 |
| Benzaldehyde | 2.5 |
| Phenol | 7.5 |
| Styrene | 0.9 |
| Trimethyl-benzene | 1.6 |
| Higher aromatics | 3.8 |
| Total | 39.2 |

In this work a resolution of T63L31ECMWF was used, which corresponds to a horizontal resolution of $1.875° \times 1.875°$ and a vertical resolution of 31 hybrid-pressure levels, extending up to the tropopause. The simulated period covers the years 2004–2005, with the first year being used as spin-up, and the year 2005 being used for the analysis. The feedback between radiation and chemistry was decoupled to avoid any influence of chemistry on the dynamics (QCTM mode (Deckert et al., 2011)). As a consequence, every simulation discussed here has identical meteorology (i.e. binary identical transport).

To analyze the influence of aromatic compounds on atmospheric composition, we performed a comparison between two scenarios. The baseline scenario, called *REF* scenario, excludes the emissions of aromatic compounds. The second scenario, called *AROM* scenario, includes all emissions from anthropogenic, biogenic, and biomass burning sources of the following aromatic compounds: benzene, toluene, xylenes (lumped), phenol, styrene, ethylbenzene, trimethylbenzenes (lumped), benzaldehydes, and higher aromatics (as representative of aromatics with more than nine carbon atoms). Both scenarios are identical aside from emissions (in the baseline case there is no chemistry of aromatic species).

We used the Representative Concentration Pathways (RCP) inventory for anthropogenic emissions (van Vuuren et al., 2011), distributed vertically as in (Pozzer et al., 2009); the MEGAN model for biogenic emissions, (Guenther et al., 2012), and the MESSy submodel Bioburn—which integrates the Global Fire Assimilation system (GFAS) inventory (Kaiser et al., 2012)—for biomass burning. In the *AROM* scenario, emissions are identical to those in Cabrera-Perez et al. (2016), except for the emissions of *higher aromatics*. In this work, emissions of *higher aromatics* were expanded to include biomass burning emissions. Emissions of aromatics sum to 35 TgC/yr, of which 3.4




TgC/yr are higher aromatics (the details of the emission factors used can be found in the supplement doi:10.5194/acp-0-1-2017-supplement). The atmospheric oxidation of aromatic compounds is performed by the MECCA sub-model (Sander et al., 2011). The complete description of the model setup—including emissions, the chemical mechanism used, and the evaluation of the *AROM* scenario—are included in Cabrera-Perez et al. (2016).

The products from the oxidation of aromatic compounds have reduced volatility, allowing them to partition into the aerosol phase and form SOA. This removal process of aromatic trace gases can significantly reduce the mixing ratios of the aromatic oxidation products. Since SOA formation is outside the scope of this work, additional channels in the chemical mechanism have been added to account for loss via SOA formation after the first oxidation step, using the yields from Ng et al. (2007). The modifications in the chemical mechanism are described in the supplement (doi:10.5194/acp-0-1-2017-supplement). This approach avoids a possible overestimation of atmospheric concentrations of aromatic oxidation products but simultaneously limits the possible impact of aromatics products on the gas-phase chemistry.

## 3  Results/ Results discussion

Figure 2 shows the annual average mixing ratios of the sum of all aromatic compounds included in the numerical simulation. The mixing ratios are higher in continental areas and close to the surface. The highest mixing ratios are found in East and South Asia, as well as in parts of Europe and the US, reaching up to ppb levels. The background mean mixing ratios in oceanic areas of the Southern Hemisphere are on the order of a few ppt.

In this section we compare the *AROM* scenario to the baseline *REF* scenario (i.e. *AROM-REF*).

### 3.1  Hydroxyl radical (OH)

Figure 3 (upper right) shows the relative difference (in %) in annually averaged daytime OH surface concentrations between the *REF* and *AROM* scenarios. Eastern Asia, Europe, and the east coast of the US show increased OH concentrations. The fractional increase in OH is up to 8% in Eastern Asia, 6% in Europe, and 4% in the US. Comparing the OH concentrations (Figure 3, upper left) with the relative and absolute differences, we notice that regions with the maximum (relative and absolute) difference (Figure 3, upper right and bottom left) match the territories where OH is quickly recycled in the atmosphere, as a result of the large anthropogenic VOC emissions in the presence of $NO_x$. Everywhere else, we find a net depletion of OH. Indonesia and its surrounding areas, and continental areas on the equatorial belt both have the strongest OH depletion. During the day in the northern hemisphere the relative differences of the annual zonal mean (Fig. 3, bottom right) globally decreases, with a (negative) gradient towards the free troposphere, reaching the maximum decrease at 200–300 hPa.





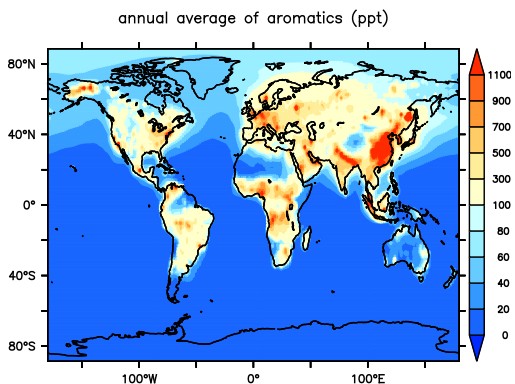

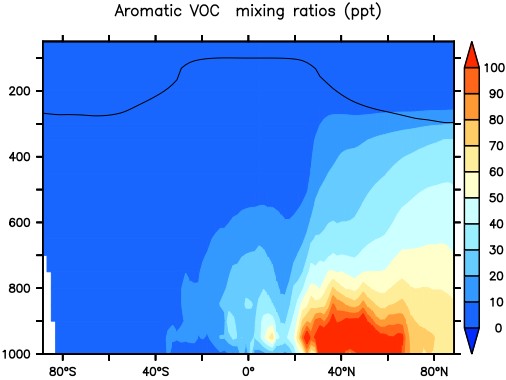

**Figure 2.** Top, annual mean of surface mixing ratios of the sum of aromatics. Bottom, zonal annual mean. Units are ppt.

155      During the night, an increase up to 6% is found between 800–600 hPa in the northern hemisphere. At the surface, annual average OH concentrations decrease in Europe, the US and Eastern Asia due to its reactions with organics. The largest absolute differences are a decrease of $1 \times 10^3$ mlc/cm$^3$ in India and Central Africa (approximately 6–10%). Nevertheless, due to the limited oxidation capacity and influence of OH–on CH$_4$ and CO–at night (Lelieveld et al., 2016), we consider that these

160     changes are of less relevance than the changes found during the day.

     The maximum instantaneous OH differences at the surface during the day exceeds 100% in the northern hemisphere continental areas during the winter, spring, and fall seasons, with absolute differences larger than $20 \times 10^5$ mlc/cm$^3$ in Eastern China and Central Africa.




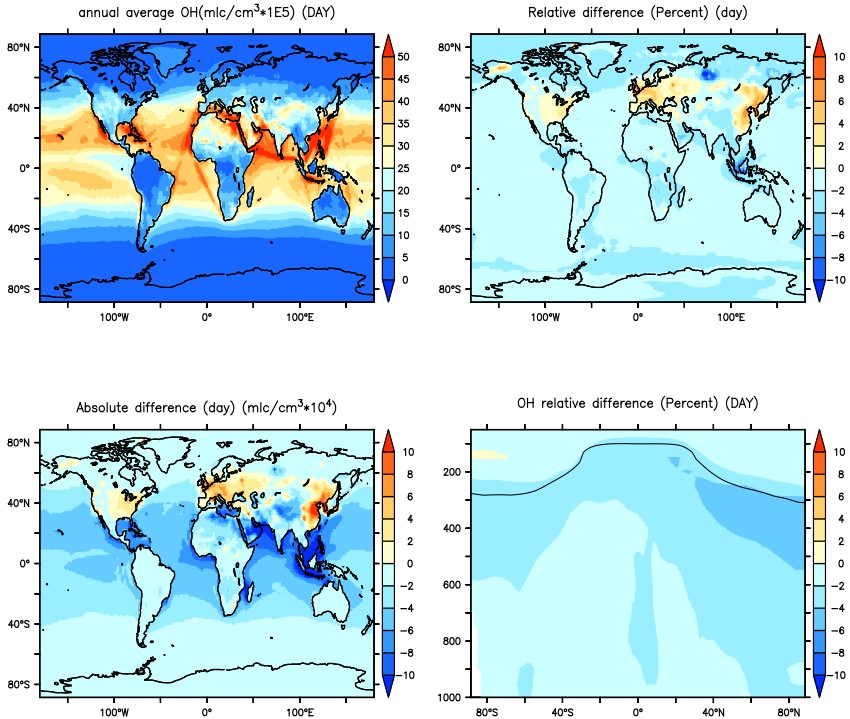

**Figure 3.** On the upper left panel, annual average surface concentrations of OH during day time (*REF* scenario). On the upper right, surface OH relative difference between aromatic and no-aromatic scenarios expressed in %. On the lower left, OH absolute difference between mentioned scenarios. On the lower right, zonal relative differences (in %).

On the seasonal level, higher OH concentrations are found over continental areas during the winter and spring than in summer and autumn (see doi:10.5194/acp-0-1-2017-supplement). During the daytime in northern hemisphere winter the relative increase in OH exceeds 50% (e.g. China and Europe). These differences correspond to absolute increases of approximately $20 \times 10^4$ mlc/cm$^3$ in China and West Asia, and of less than $6 \times 10^4$ mlc/cm$^3$ in the US and, Europe. In summer, Europe is the only region where OH concentrations increase when aromatics are included, although not exceeding 5%. The increase of OH production via $NO + RO_2$ through the increase of $RO_2$ concentrations are due to aromatics oxidation. Although there is a ubiquitous decrease in $NO_x$, this does not seem to limit OH formation. In the southern hemisphere and in oceanic areas the net effect of introducing aromatic compounds into the system results in a net depletion of OH due to the increase of CO as final product of the oxidation scheme (see Sect. 3.5.3). Figure 5 shows the seasonal cycle



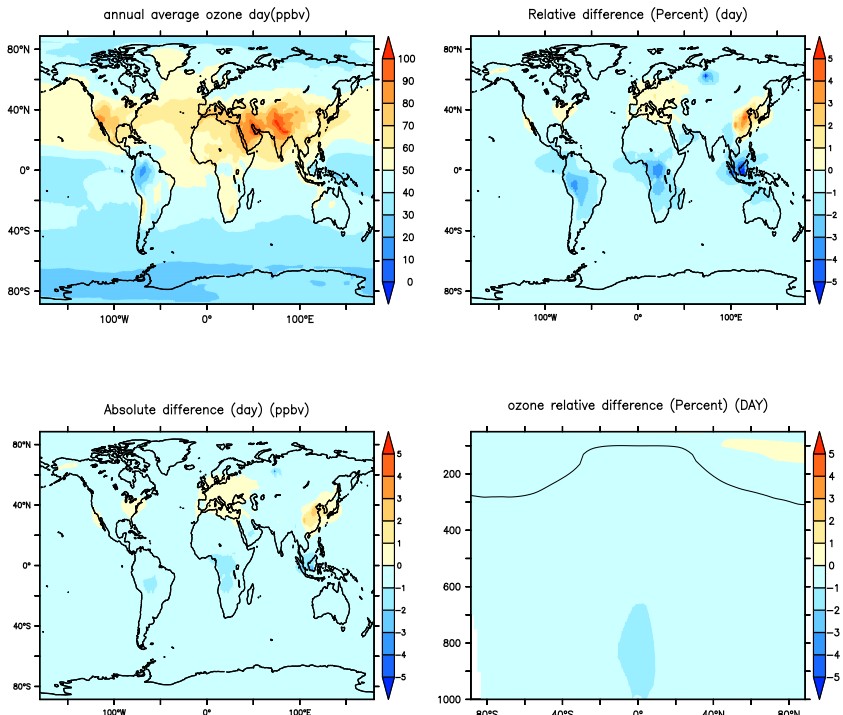

**Figure 4.** On the upper left panel, annual average surface mixing ratios of $O_3$ during day time (*REF* scenario). On the upper right, surface OH relative difference between aromatic and no-aromatic scenarios expressed in %. On the lower left, OH absolute difference between mentioned scenarios. On the lower right, zonal relative differences (in %).

175    of the relative difference between both scenarios for the northern and southern hemispheres. We find that the relative difference varies between -3.5% for winter months up to -1.5% in summer months. The cycle is reversed for the southern hemisphere, where the changes in OH concentrations are due to transported species (mostly CO), making changes relatively homogeneous.

Table 2 lists the OH concentrations for the *REF* and *AROM* scenarios, and the relative and abso-
180    lute difference averaged globally in the boundary layer. We obtain an enhancement of 2–3% in the atmospheric lifetime of methane, along with a decrease in OH concentration. Methane is a long-lived species whose breakdown is driven by OH (Crutzen and Zimmermann, 1991); its lifetime is a measure of the OH concentration (Prather and Spivakovsky, 1990). The decrease in the OH concentration thus explains methane's increased lifetime. OH concentrations are also in line with
185    values estimated by Lawrence et al. (2001), although estimated methane lifetimes are shorter, as



these lifetimes were estimated only for the boundary layer. Tropospheric methane lifetimes were close to literature estimates: The *REF* scenario yields a methane lifetime of 9.74 years, and *AROM* a methane lifetime of 10.05 years, however, this does not include $CH_4$ loss in the stratosphere and from soil uptake, which reduce the overall lifetime of methane. Literature estimates vary between 7.1 and 10.6 years (Voulgarakis et al., 2013; Lawrence et al., 2001). Separating day and nighttime, the the OH concentration decrease approximately 2% and 1% –independent of the weighting method–, respectively.

The changes in OH concentrations are dependent on to the $NO_x$ regimes, in combination with increases in CO concentrations and changes in VOC mixing ratios as a consequence of aromatic oxidation. In areas with high $NO_x$ and VOC mixing ratios, ozone is formed, which under photolysis produces OH (R5-R6 ). In contrast, in regions with low $NO_x$ concentrations or absent anthropogenic emissions OH recycling is limited by $NO_x$, forcing the aromatic oxidation products to react with $HO_2$ or peroxy radicals ($RO_2$). At the same time, CO is formed from the oxidation of aromatics (its main sink is reaction with OH), strengthening the removal of OH. Moreover, the oxidation chain of aromatics includes a large number of reactions involving OH. Therefore, positive changes in OH concentrations take place in continental areas with large $NO_x$ emissions, unlike, for example oceanic areas, where $NO_x$ concentrations are low.

For $HO_2$ a relative decrease of less than 1% in the atmospheric global burden is found. At the regional scale, only Europe and East Asia have relative increases in surface mixing ratios by more than 10%. The oxidation of carbon monoxide by OH is the main source of $HO_2$ (Lightfoot et al., 1992), serving as a buffer for OH; in addition, aromatic chemistry contains a large number of reactions leading to formation of $HO_2$ (e.g. the initial reactions of benzene + OH leads to Phenol + $HO_2$). The updated benzaldehyde photolysis used in this study was also more efficient than the one previously used, making it a significant source of $HO_2$ radicals. The small impact of aromatics observed on the $HO_x$ budget was expected, since this budget is well buffered against perturbations (Montzka et al., 2011; Lelieveld et al., 2016).

Contrary to expectations, we found large increases of HONO mixing ratios in continental areas, generally with the sign of this change opposite to that of OH. These relative changes reached more than 50%, specifically in Africa, South-America, the Arabian Peninsula and South-East Asia. The reason for this pattern is explained by the photolysis of nitrophenols, which leads to HONO formation (Bejan et al., 2006; Cheng et al., 2009). In the *AROM* scenario, approximately 5.7% of the HONO formation is directly related to nitrophenol photolysis at surface. Nevertheless, the net effect on the atmospheric burden is a depletion of around 1.5%.

### 3.2 Ozone

Differences between *AROM* and *REF* of daytime annual averages of surface ozone mixing ratios are shown in Fig. 4 (upper right and bottom left). In general, surface ozone depletion occurs over large





**Table 2.** Estimated global averaged OH concentrations in the boundary layer by three different approximations Lawrence et al. (2001) and global lifetimes of methane. Calculations for the reference scenario and the aromatics scenario and the differences between them.

| | Lifetime $CH_4$ (years) | OH concentration (VOL) x $10^6 mlc/cm^3$ | OH concentration (MASS) x$10^6 mlc/cm^3$ | OH concentration (CH4) x$10^6 mlc/cm^3$ |
|---|---|---|---|---|
| **No aromatics** | | | | |
| Global | 4.45 | 1.30 | 1.27 | 1.46 |
| Global day | 2.46 | 2.33 | 2.29 | 2.58 |
| Global night | 236 | 0.03 | 0.02 | 0.03 |
| **Aromatics** | | | | |
| Global | 4.53 | 1.28 | 1.25 | 1.43 |
| Global day | 2.51 | 2.29 | 2.24 | 2.53 |
| Global night | 235 | 0.03 | 0.02 | 0.03 |
| **Relative Difference (%)** | | | | |
| Global | 1.91 | -1.87 | -1.86 | -1.88 |
| Global day | 1.95 | -1.89 | -1.88 | -1.90 |
| Global night | 0.21 | 0.51 | 0.56 | .20 |
| **Absolute Difference ($10^4 mlc/cm^3$)** | | | | |
| Global | 0.08 | -2.42 | -2.37 | -4.10 |
| Global day | 0.05 | -4.40 | -4.30 | -7.25 |
| Global night | -0.50 | 0.01 | 0.01 | <0.01 |

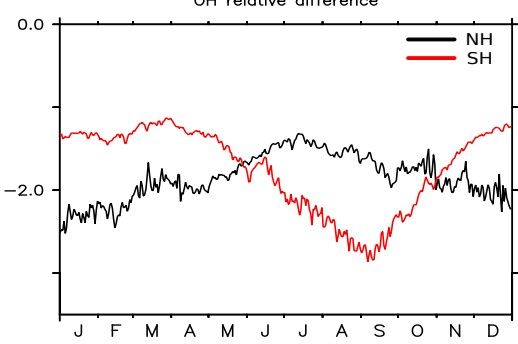

**Figure 5.** OH relative difference (expressed in percent) in the boundary layer between the *AROM* and *REF* scenarios. In black, values for the northern hemisphere. In red, values for the southern hemisphere.





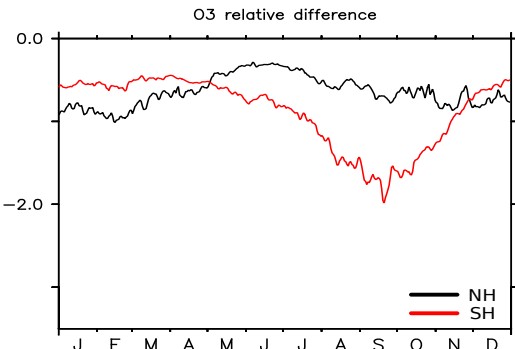

**Figure 6.** Same as in Fig. 5 for ozone.

**Table 3.** Estimated global averaged ozone mixing ratios in the boundary layer Lawrence et al. (2001). Calculations for the reference and aromatic scenarios.

| | $O_3$ mixing ratio ppt |
|---|---|
| **No aromatics** | |
| Global | 1.11 |
| Global day | 1.05 |
| Global night | 1.18 |
| **Aromatics** | |
| Global | 1.10 |
| Global day | 1.05 |
| Global night | 1.17 |
| **Relative Difference (%)** | |
| Global | -0.72 |
| Global day | -0.57 |
| Global night | -1.76 |
| **Absolute Difference (ppt)** | |
| Global | -0.69 |
| Global day | -0.63 |
| Global night | -0.78 |

areas of the globe. Maximum depletion occurs in equatorial areas, of between $10°N$–$20°S$ (down to 4%, a drop of approximately 5 ppbv). In contrast, some regions of Europe and China have small increases (less than 5%); these areas match the regions with the largest $NO_x$ annual mean mixing



ratios. In Central African and Indonesian territories, where strong biomass burning takes place (i.e. large emissions of VOCs), significant decreases in ozone formation are observed.

No significant differences between daytime and nighttime surface ozone mixing ratios were found, due to the relatively long lifetime of ozone (about three weeks), which is several times longer than the lifetime of aromatics (Stevenson et al., 2006). On the other hand, larger relative differences at the

surface are found during part of the year. During the winter months, a relative increase of more than 10% is found in China, and of 1–4% in Europe and the US. China had increased ozone mixing ratios for almost the entire year (excluding the summer months), and ozone depletion in central Africa remains relatively constant year-round. During the summer and fall seasons, the strongest depletion is observed in the southern hemisphere (approximately 7%).

The seasonal distribution of the relative differences (Fig. 6) shows lower amplitude than for OH, but similar patterns. In the northern hemisphere, greater relative differences occurred between scenarios in the winter months than in the summer months; in the southern hemisphere this pattern is reversed. The relative differences range within 1%–2%. In the southern hemisphere a maximum difference of 2% is found over biomass burning activity regions.

Maximum increases of ozone of more than 10% occur over the northern hemisphere. In the US, Europe, and China ozone mixing ratios can increase by more than 20%. Large peaks were also observed in Central Africa, due to strong biomass-burning events.

Relative changes in tropospheric ozone were found to be homogeneous within the northern hemisphere (Fig. 4, bottom right). There did not appear to be any differences between day and night,

with decreases below 2% in the boundary layer and lower part of the free troposphere (up to 7 km). Above 7km height, the relative differences decrease.

Table 3 presents annual global mean ozone mixing ratios in the boundary layer for both scenarios—weighted by two different methods, but only mass weighted is shown, as both methods lead to the same results—as well as the relative and absolute differences. Both methods show good agree-

ment in the mixing ratios. Relative differences are approximately 1%. During nighttime, relative differences increase to 1.8%. Compared to OH, relative differences are lower for ozone, suggesting that ozone chemistry is less affected by aromatics than OH.

The simulation results show a weakening in the ozone formation; this result brings an interestingly mismatch with former studies (e.g.Butler et al. (2011)). Aromatics, especially toluene and xylenes,

have significant ozone formation potentials among VOCs. It could therefore be expected an overall increase in ozone formation when aromatics are introduced into the system. However, there are important differences between the work by (Butler et al., 2011) and this work; the first is based on a box model study for two cities, and the current work focuses on the global scale. It is not clear whether the first work takes into account the dry deposition and scavenging, which can be

important factors. Last, the capability of the global model to represent correctly the chemistry in cities is limited due to the coarse model resolution. The ozone depletion in the this model study





is due to (i) the decrease in $NO_x$ mixing ratios –limiting ozone formation–, and to (ii) increasing radical production (OH, $HO_2$, and $RO_2$) in ozone-depleting regimes, which enhances reactions of $O_3$ with $HO_2$ and OH. Growth in ozone mixing ratios is observed in regions of high $NO_x$ mixing 265 ratios, where the limiting factor for ozone formation is hydrocarbon mixing ratios.

### 3.3 $NO_x$

The simulated annual mean $NO_x$ mixing ratios at surface were slightly lower in the *AROM* scenario than in the *REF* scenario at surface. One reason of decreased $NO_x$ is the addition of chemical species containing nitrogen (e.g. nitrophenols) in the *AROM* scenario, thereby transferring part of the $NO_x$ 270 burden to the nitrogenated species. Thus, aromatic nitrogenated species become intermediates for the deposition of nitrogen.

The comparison between the *AROM* and the *REF* model results shows significant $NO_x$ depletion in central Africa, the Amazon forest, China, and Indonesia, with relative differences reaching approximately 10% (Fig. 7). In Europe, the US, and the Arabian Peninsula, reductions in $NO_x$ mixing 275 ratios did not exceed 5%. Oceanic areas show small decreases, although $NO_x$ mixing ratios are several orders of magnitude lower than over continental territories, with absolute changes not exceeding 1 ppt. In the boundary layer, $NO_x$ mixing ratios decreased by 5% in the *AROM* scenario.

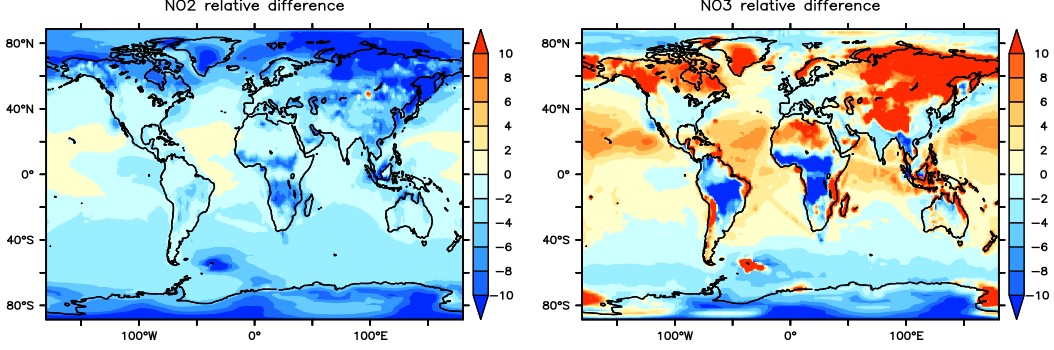

**Figure 7.** Same as in Fig. 5 for $NO_2$ (left) and $NO_3$ (right).

### 3.4 $NO_3$ and $HNO_3$

For annual average boundary layer $NO_3$ mixing ratios we find a global increase of 3% during night- 280 time and 6% during the daytime.

At the regional scale, the largest daytime changes occur in northern Africa, the Arabian Peninsula, and northern Asia, with increases of 20–30%. An increase of more than 30% was also observed over



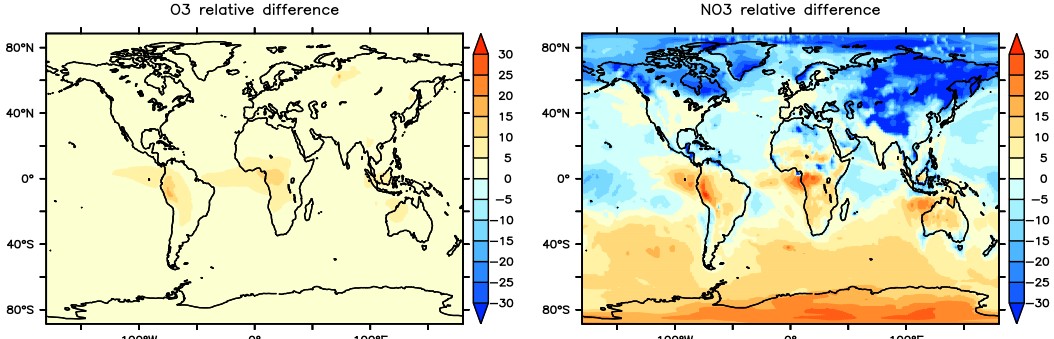

**Figure 8.** Relative difference ((*No phenoxy channel-AROM*) / *AROM*) expressed as a percent for ozone (left) and $NO_3$ (right).

the Tibetan plateau, although mixing ratios in this area are generally low. Decreases of up to 10% were observed in central Africa and in the Amazonian areas. Since $NO_3$ concentrations are low

during daytime, these changes do not significantly affect atmospheric chemistry.

During nighttime, the difference between the scenarios are larger, with a 30% decrease in mixing ratios in central Africa and the Amazonian areas, an increase of more than 30% in the Tibetan plateau and northern Asia, and no change in northern Africa.

The net formation or depletion is defined by the competition between the amount of aromatic

products consuming $NO_3$ versus the strength of the phenylperoxy channel, which leads to $NO_3$ formation. We investigated the importance of the phenylperoxy channel for $O_3$ and $NO_3$ mixing ratios. The different channels, based on the work of Jagiella and Zabel (2007), were added to the current mechanism (Cabrera-Perez et al., 2016). Figure 8 shows the comparison for $O_3$ and $NO_3$ between the *AROM* scenario and an identical scenario without the phenoxy radical channels. Ozone mixing

ratios increased as a result of neglecting the channel that transforms $NO_2$ into $NO_3$, and more $NO_2$ is therefore available for the catalytic process leading to ozone formation. The phenylperoxy channels quickly convert $NO_2$ into $NO_3$ (e.g. $C_6H_5O_2 + NO_2$ leads to $C_6H_5O + NO_3$ in the mechanism). For $NO_3$ the areas with increases in the mixing ratios can be explained by the increases in ozone and $NO_2$ mixing ratios, leading to an increase in the $NO_3$ formation. These increases reach up to 30%

in equatorial regions. In contrast, there are regions showing decreases by more than 30% in $NO_3$ mixing ratios; these decreases suggest a large strength of the phenylperoxy channels in certain areas of the northern hemisphere.

For $HNO_3$, the atmospheric burden decreases by 1% in the *AROM* scenario. However, surface mixing ratios showed increases up to 20% in East and South-East Asia, Central Africa and South



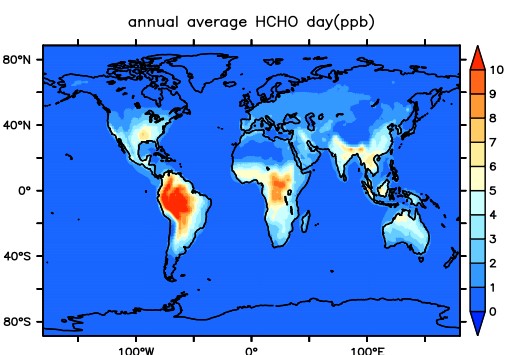

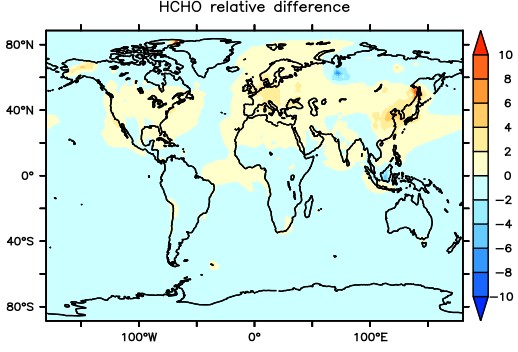

**Figure 9.** Top. Annual average surface mixing ratios of HCHO in ppt. Bottom, relative difference in %.

America. In Europe and the US, surface mixing ratios increased up to 6%. In oceanic areas changes remain below 2%. The increases can be explained by the number of reactions leading to $HNO_3$ formation (e.g. the reactions of xylenes, trimethyl benzenes or ethylbenzene with $NO_3$ form $HNO_3$).

### 3.5 VOC

#### 3.5.1 Formaldehyde

The main photochemical source of formaldehyde in the background troposphere is methane oxidation; in continental areas, VOC (including aromatic compounds) oxidation is the main source, and its main sink is reactions with OH. Comparing our two scenarios, we find a depletion in the formaldehyde surface mixing ratios (on an annual basis) in the Amazonian and central African regions—two areas that typically have higher formaldehyde mixing ratios (Figure 9). In contrast, we observed in-





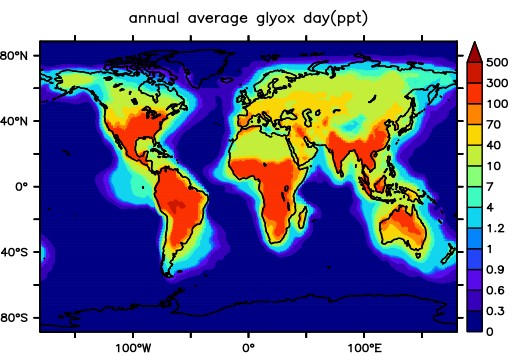

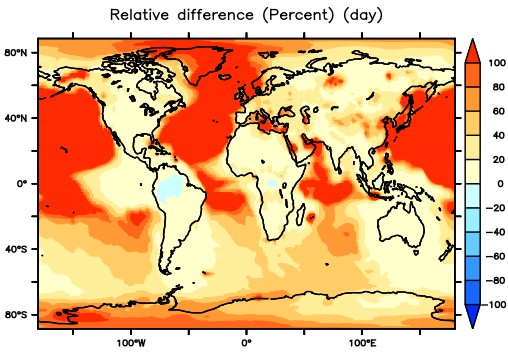

**Figure 10.** Top. Annual average surface mixing ratios of glyoxal in ppt. Bottom, relative difference in %

creases of up to 6% in these mixing ratios in China and Europe. On the global level there is a decrease
in the atmospheric burden of formaldehyde of 1.6%. These changes in the formaldehyde distribution
can be explained by changes in the OH mixing ratios, as well as the formaldehyde production from
the aromatic oxidation. For instance, large urban areas have high levels of aromatics, which leads
to increases in formaldehyde formation. In regions where aromatics deplete OH, methane and other

VOCs then form less formaldehyde.

### 3.5.2   Glyoxal

With respect to glyoxal, we find a global increase of 20% of the simulated mixing ratios at the
surface. At the regional scale, China has the strongest absolute difference, with an increase of ap-
proximately 70 ppt (80%); followed by India and the Arabian Peninsula, where increases rise by

more than 40 ppt. In Figure 10 (bottom), increases larger than 100% can be seen in oceanic ar-



eas; however, this large relative change is associated with very low mixing ratios (less than 1 ppt). In continental areas over the northern hemisphere, increases of more than 10% are found. Relative increases are lower in the southern hemisphere than in the northern hemisphere, because the main source of glyoxal in the southern hemisphere is isoprene (Fu et al., 2008). Only in some regions

of Africa and the Amazon a decrease (less than 2%) was observed; this was caused by a depletion of OH, and by isoprene being almost the only source of glyoxal. The atmospheric burden of glyoxal is subject to a net increase of approximately 10%. Nevertheless, the burden (62 Gg in the *REF* scenario) seems to be much larger than in Fu et al. (2008); Myriokefalitakis et al. (2008) (15–20 Gg). This discrepancy is attributed to the missing mechanism of SOA formation, which has been

estimated to account for approximately 20% of the total atmospheric sink (Stavrakou et al., 2009).

### 3.5.3    Carbon monoxide

The reaction chain in the oxidative process of aromatics produces carbon monoxide (CO), a relatively long-lived molecule (1–2 months). CO can travel long distances from its source, although this lifetime is not long enough to allow it to cross hemispheres (Daniel and Solomon, 1998). CO

mixing ratios generally increase on the global scale, indicating a small addition to the carbon budget. When comparing both scenarios, we find an increase of 3% in the atmospheric burden of CO, which corresponds to an increase of 14 Tg. The CO burden estimated by the model in the *REF* scenario is 546 Tg.

## 4    Sources of uncertainty

There are a number of sources of uncertainty in the estimation of the impact of aromatics on tropospheric chemistry. Firstly, emissions of aromatics have high uncertainties. One reason is that few databases provide anthropogenic speciation of aromatic VOCs. In the case of the RCP database (van Vuuren et al., 2008), speciation entails fractioning the total VOC flux into different species (a top-down approach) (Moss et al., 2008). In the case of biomass burning emissions, information

regarding the emission factors for each vegetation type is not always available for all species, and when available the variability is large (e.g. emission factor for tropical forest emissions of benzene is $0.39 \pm 0.16$ g/kg). With regard to biogenic emissions in the present study, only toluene was included, despite that this source is of minor relevance as compared to other sources. The emissions fluxes calculated by the MEGAN model are strongly dependent on input data: e.g., on temperature

or incoming radiation (Guenther et al., 2012). The model calculates a difference of 15% in the total emissions between the T42 and T106 resolutions. Interannual variability of toluene is estimated to be 6% of the total annual emissions (Sindelarova et al., 2014).

Atmospheric changes produced by aromatics are strongly dependent on $NO_x$ mixing ratios. The model tendency to underestimate $NO_x$ mixing ratios is small (Jöckel et al., 2006), but this is a





possible source for underestimating ozone formation. Finally, it must be stressed that the EMAC
       model overestimates ozone mixing ratios (Jöckel et al., 2006).

       Another source of error is due to the chemical oxidation mechanism (based on MCMv3.1), which
       in general overestimates peak ozone mixing ratios (e.g. by more than 15% in the case of toluene), and
       underestimates OH formation (up to 80%) and NO oxidation rates Bloss et al. (2005). In the case of
OH, our version of the chemical mechanism includes HONO formation channels from nitrophenol
       photolysis, which contributes to OH formation during daytime.

       In the mechanism used in this work, the oxidation of benzene and toluene was taken from MCM.
       For the rest of the aromatics that were included, the second oxidation products are directly linked to
       those of toluene. This approximation implies a less accurate representation of the oxidation of these
species. Consequently, a comparison of simulation results with observations show relatively good
       agreement for benzene and toluene, although the model has difficulty reproducing xylene mixing
       ratios (Cabrera-Perez et al., 2016).

       The largest uncertainty was generated by the treatment of SOA formation. Although approxi-
       mately 23% of the aromatics emissions lead to SOA (similar to the 18% estimated by Henze et al.
(2008), there are uncertainties with regards the use of the yields from Ng et al. (2007) to simulate
       real atmospheric conditions, as discussed in Henze et al. (2008). We consider it to be a plausible
       assumption that the channels for SOA formation occur after the first oxidation step, although this
       assumption implies that all secondary reactions are equally affected. However, it is hard to estimate
       whether this methodological choice leads to an underestimation or overestimation of the effects of
aromatic compounds on atmospheric species.

       ## 5   Summary

       This work investigated the effect of aromatic compounds on the chemistry of the troposphere at the
       global scale, with the help of the global circulation atmospheric chemistry model EMAC. A baseline
       case scenario and a sensitivity run were compared, with the first excluding aromatic compound
emissions and second including them. To accurately describe the oxidation chain of the species
       involved (all simple monocycle aromatic compounds) we used a detailed chemical mechanism based
       on MCM.

       At the global scale, OH concentrations decrease by 2–3% once aromatics are included. On the
       regional scale, areas with high levels of aromatics have decreases of more than 10%, while regions
with large $NO_x$ mixing ratios show increases of up to 10%. This decrease in OH mixing ratios can
       alter the VOC distribution. For example, the formaldehyde atmospheric burden decreases by 1.2%.
       For ozone similar results to those of OH are found, with a global net decrease of 1%. However, the
       relative importance of aromatics at the regional scale can cause increases or decreases of more than
       10% during the winter season.



When aromatic compounds are included, increases in carbon-containing compound emissions add to the carbon monoxide burden. In the case of glyoxal, an increase of 20% in the mixing ratios was calculated. This increase in the carbon-containing compounds explains the net decrease in OH concentrations over remote regions.

    An important outcome is that aromatics have a larger impact in the $NO_x$ than the $HO_x$ budget. 400  $NO_2$ is depleted due to reaction of nitrogen compounds with aromatics, with this nitrogen being deposited more efficiently. In contrast, the $NO_3$ atmospheric burden increases by 6%, with changes by more than 30% in several areas of the northern hemisphere. Aromatics can thus be important for nighttime chemistry. The depletion of $NO_x$ in combination with the increase in CO leads to OH depletion.

We conclude that, although aromatic compound chemistry at the global scale has a relatively minor impact, at a regional scale, the influence of aromatics in the troposphere can be important, and can be responsible for relatively large changes in OH, ozone, and especially glyoxal and $NO_3$. We therefore recommend including detailed aromatic chemistry in regional and global model simulations.

*Acknowledgements.* The authors want to acknowledge the use of the Ferret program for analysis and graphics in this paper. Ferret is a product of NOAA's Pacific Marine Environmental Laboratory (information is available at http://www.ferret.noaa.gov).



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
