# Peer review of "Global impact of monocyclic aromatics on tropospheric composition"

_Atmospheric Chemistry and Physics, 2017_

## Referee Comment (RC1) · Anonymous Referee #1 · 2 Jan 2018

The manuscript by Cabrera-Perez et al. discusses the impact of aromatic species on the tropospheric composition using the global chemical transport model MESSy/EMAC. A model simulations with aromatics included is compared to a simulation without considering aromatics and the impacts on key species such as OH, Ozone, NOx, NO3, HNO3, HCHO and glyoxal are explored. This manuscript is strongly based on Cabrera-Perez (2016), which describes the aromatic chemistry added to EMAC and evaluates the global distribution of modeled aromatic species. While I acknowledge the importance of studying how aromatics impact the chemistry of the troposphere, I see limited scientific advancement in the presented study. What is it that really can be learned from this work aside from that aromatics play a role in tropospheric chemistry? For this reason and because the manuscript rather represents a model description, I

see it better suited for a journal like GMD. Major Comments: Aromatics come from a number of different source types including both anthropogenic and natural sources. Simply turning off all aromatic species in the model independent of their origin or their reactivity does provide little practical insight. The study would provide more value if it were to look at individual source types and/or individual aromatic species. The study is conducted at a rather coarse horizontal model resolution (1.875 degrees). At this resolution urban chemistry cannot be represented, yet aromatics play an important role specifically in urban regimes. The authors themselves state in Line 260 that the coarse resolution limits the representation of urban chemistry. So why was the study not done at higher model resolution or with a higher resolution regional model? Has the aromatic chemistry in EMAC been compared to a fully explicit chemical scheme? If so, how well does it represent the full chemistry? The authors list a large range of uncertainties that impact the presented results and reduce confidence in the value of the conclusions. I agree that emissions in general can pose, in parts, large uncertainties. Have the authors looked at other inventories (HTAP, RETRO, NEIs, . . .)? What was the reason for selecting IPCC emissions? What is the reason for the high ozone bias in ECAM? Could the chemical scheme be a cause? Line 362: Do the authors imply that MCMV3.1 is incorrectly representing ozone chemistry or that the lumped chemical scheme that was developed based on MCMv3.1 and used in the current study has problems? If so, how much can one trust the derived sensitivity of tropospheric chemistry to aromatics? Does including HONO help with this problem? Has this been tested in a box model? Given the large role of SOA in aromatic chemistry, should such a study not have been conducted with a more detailed aerosol scheme that represents SOA formation. There is zero evaluation of the model with observations. Aromatic species themselves have been evaluated in Cabrera-Perez (2016), but what about e.g. $O_3$ or CO?

Minor Comments: Line 55 (and also Line 195): This is a very simplified description of ozone chemistry. High $NO_x$ regimes do not equate net ozone formation as at very high $NO_x$ concentrations ozone can actually be lost. The actual urban chemistry cannot be resolved at the coarse model resolution applied in here.

Line 125: Given how closely this manuscript is linked to Cabrera-Perez (2016) I suggest referencing some of the major findings of this study as they apply to this work.

Figure 5 and Figure 6: Is this daytime? Daytime and Nighttime? Most Figure Captions require more details on what is shown.

The authors discuss the role of a number of intermediate species. I suggest adding the chemical reactions for the aromatics as included in MESSy in the Supplement or at least include a reference to where the chemical reactions can be looked up.
* * *

---

## Referee Comment (RC2) · Anonymous Referee #2 · 12 Jan 2018

In this study, the authors have applied a global chemistry-climate model (in a chemical transport model mode) to study the impact of monocyclic aromatic species on tropospheric gas-phase composition. They assess the changes in the distribution of several air quality and climate relevant chemical species, including OH, O3,NOx, HCHO, glyoxal and CO as a result of including aromatics in the chemical mechanism. The authors find that the impact of aromatics is small on a global scale but have a bigger impact regionally. Despite finding a small impact on a global scale, the authors recommend that global models include aromatics. I do not think that the authors make a convincing case for this. Moreover several global models already include aromatics (e.g., GEOS-chem, MOZART-4). The paper in its current form does not advance our scientific understanding of the role of aromatics in chemistry and climate. Specific comments are provided

to improve the paper:

P6, section 3: Comparison with observations is important to build confidence in the simulation of aromatics by the model. How do we believe the model is simulating aromatics in the right place for the right reasons? Are there any comparisons with observations? P5,L101: Why 2004-2005 time period was chosen? Why not a more recent period as I would imagine that emissions have evolved since 2005. P5, L114-115: Which RCP emissions are being used here? P6, L146: Despite high levels of surface aromatics simulated over India and central Africa in the AROM run, why is there not a concomitant increase in day time OH over the highly polluted northern Indo-Gangetic plains or central Africa between REF and AROM? P7, L155: Where are these results shown? Figure 3. How significant are the differences, especially the small numbers, shown here. Presumably the black line on the lower right plot shows the tropopause level, please indicate this in the caption. P8,L171: Please clarify the statement "Although there is ubiquitous decrease in NOx this does not seem to limit OH formation." P8, L172-174: How significant are the decreases in the remote southern hemisphere oceanic regions? Figure 2 shows zero concentrations of aromatics in this region and given their short lifetimes I do not think they are being transported into this region? Possibly CO is being transported. If so, can you please provide a plot of changes in CO distribution with a significance estimate. P8,L174: Figure 5 should be moved to Figure 4 if it is being discussed before the existing Figure 4. P9, Table 2: Please clarify in the caption that the global methane lifetime refers to lifetime of $CH_4$ in the global boundary layer. Also suggest explicitly showing how this lifetime is calculated. In particular, how is the boundary layer diagnosed? P10, L187-188: How do the tropospheric methane lifetime compare against those from other models, for example the ACCMIP models for year 2000? P10, L195: High NOx can also result in titration of ozone. P10, L203-218: Please point to figures in the supplementary in this discussion. P 9, Figure 4: The figure shows O3 changes while the caption mentions OH changes. Please correct the caption. P13, L259: Is deposition of individual aromatics considered in this study? If so, how are the dry deposition velocities calculated? P18,L348: The van Vuuren et al

reference is wrong.

---

## Referee Comment (RC3) · Anonymous Referee #3 · 16 Jan 2018

In this work, the authors compare differences in predicted tropospheric species distributions and relative concentrations between two model simulations: (1) a base case (REF) in which no aromatics are emitted globally to (2) a case in which C6-C11 aromatics are emitted globally (AROM). Both model simulations include detailed aromatic chemistry, but since there are no aromatic emissions in REF, only AROM contains aromatic chemistry impacts. The base case (REF) is not consistent with recent, state-of-the-art models (GEOS-chem, MOZART-4, etc.), in which benzene, toluene, xylenes, or some parameterized combination of these and their chemistry is included. A comparison between a model simulation in which a subset of aromatics are parameterized as one or two or three species and then compared to the AROM simulation would be more scientifically relevant. It is likely, however that the differences (globally) of such a

model simulation would be insignificant, as the differences in the AROM-REF situation presented here have very little significant globally. The other issue is that the model resolution is very coarse, which is also probably why there isn't a significant impact from the addition of the larger, more reactive aromatics – they're simply lost into such a coarse grid.

In general, the comparisons shown simply reinforce that it is worthwhile to include aromatics and aromatic chemistry in global models, but this is not at all novel as aromatic chemistry has been included in global models for years.

A comparison of a reasonable current chemistry scheme with emissions of a subset of aromatics compared to the full aromatic scheme and expanded aromatics emissions using a finer model grid over a regional scale may be more appropriate to look at the impacts of including larger aromatics (C9+ aromatics) on urban and regional scales, where the reactive aromatics likely have a more significant impact on tropospheric chemistry. This would be a more novel and valuable approach.

Throughout the paper, and in particular the introduction, there are awkwardly-worded sentences that would significantly benefit from having a careful once-over/proofread/edit by a native or strong English speaker. E.g., Page 1, line 20; Page 3, lines 63-64; Page 4, line 74; Page 6, line 161; Page 10, line 193; Page 11, Table 2 title.

Specific comments

Page 1, line 5 – in referring to the inclusion of "aromatic compounds" in the abstract, perhaps state which species and/or class of aromatics are included, and to what degree their chemistry is included. Even stating C6-C11 aromatics would be helpful.

Page 1, line 7 – Be specific about the changes – relative or absolute. Also, here and elsewhere in the paper, the word "found" is used to refer to differences between the base REF case with no aromatic hydrocarbon emissions and the AROM case in

which aromatics emissions are included. I would argue that nothing is "found", but rather, a difference in the atmospheric burden of particular species in the simulation results was noted. "Found" implies that measurements were made, and there were no measurements made or reported in this paper. I would suggest replacing "found" with "predicted", or something similar (see also page 1, lines 9, 12, etc.)

Page 6, line 121 - It would be nice if I didn't have to go read another paper to get all the details about the additional chemistry. A summary or brief description of the detailed aromatic chemistry in Cabrera-Perez et al. 2016 would be nice to include in this paper.

Page 6, line 143 – define "daytime". Also, define "surface".

Table 1 – the title stating "This emissions are the same as in … but for higher aromatics" is both grammatically incorrect and not clear. This should be a footnote in the table, and more specific, or perhaps, to make it more clear, show the 2016 paper emissions in the table as another column. More importantly, there is an inconsistency between the table and text: the total aromatic emissions in the table are 39.3 TgC/yr of which 3.8 TgC/yr are higher aromatics, while the text (Page 5, line 120) states that it is 35 TgC/yr, of which 3.4 TgC/yr are higher aromatics. This needs to be reconciled. Also, Trimethyl-benzene should indicate "trimethylbenzenes" (there are three different trimethylbenzenes). Also, what about the ethyltoluenes?

Technical corrections

Figures - rather than saying "upper left panel", "top panel", etc., simply label the different panels of each figure a, b, c, etc., and then refer to them in the figure caption with (a), (b), (c), etc. When referring to the supplement, refer the reader to a specific Table (S1, S2, etc.) or Figure S1, S2 in the supplement, and not the document as a whole.

Page 1, line 16 – change "comprises" to "comprise".

Page 2, line 25 – RO2 is not "the peroxy radical", but rather "an organic peroxy radical".

Page 3, line 57 – change "In contrast, the high NOx…" to "In contrast, O3 in the high

NOx regime..." (It is the O3 that is limited by VOC concentration, not the high NOx regime.

Figure 2 – there is an extra space after Aromatic VOC in the "title" of the bottom panel. Also, there is inconsistency throughout the paper in the capitalization of figure titles (which is odd – typically figures do not have titles).

Page 7, line 157 and elsewhere – mlc is not an acceptable shortform for molecules. Either spell it out entirely (preferred), or use the somewhat acceptable "molec" as a shortform.

Page 9, Figure 4 caption – Internally inconsistent – refers to O3 and OH.\

Page 13, line 246 – add a space between 7 and km.

Page 13, line 254 – add a space between e.g. and Butler.

Supplement – Table (S)1. "higher" does not need to be capitalized. The sentences after the * in the table title should be in the footnotes. Also, the reference to extra-tropical forests is unclear. Define the references to PTR (naphthalene and C11 Aromatics).

Supplement Figure (S)1. Change "(Arom – base)/base, %" to "(AROM-REF)/REF, %" to be consistent with the main text, and this should really be the y-axis title, not the figure title. For the y-axis labels, use decimals instead of commas, or just use integer values.

Figure S2 – units in %?

Figures S3 and S4 – the titles are redundant with the figure captions.

---

## Author Comment (AC1) · 2 May 2018

*We thank the referee for her/his valuable comments. Here the comments are repeated with our answers.*

**The manuscript by Cabrera-Perez et al. discusses the impact of aromatic species on the tropospheric composition using the global chemical transport model MESSy/EMAC. A model simulations with aromatics included is compared to a simulation without considering aromatics and the impacts on key species such as OH,Ozone, NOx, NO3, HNO3, HCHO and glyoxal are explored. This manuscript is strongly based on Cabrera-Perez(2016), which describes the aromatic chemistry added to EMAC and evaluates the global distribution of modeled aromatic species. While I acknowledge the importance of studying how aromatics impact the chemistry of the troposphere, I see limited scientific advancement in the presented study.**

**The manuscript rather represents a model description, I see it better suited for a journal like GMD.**

*Journals like GMD are rather technical and suited for model development/evaluation. Here, impact studies are performed on a global scale and therefore, to our opinion, the topic fits perfectly within the journal's scope.*

**Major Comments: Aromatics come from a number of different source types including both anthropogenic and natural sources. Simply turning off all aromatic species in the model independent of their origin or their reactivity does provide little practical insight. The study would provide more value if it were to look at individual source types and/or individual aromatic species.**

*We thank the referee for pointing this issue. We fully agree with her/him, and to improve the manuscript, we performed simulations for studying the relevance of each source (anthropogenic, biogenic and biomass burning) and also including only benzene, toluene and the rest of compounds.*

*We performed two series of sensitivity simulations. In the first series, three simulations were run, where each of the different sources were switched off, i.e. anthropogenic, biogenic and biomass burning emissions, respectively. In the second series, four simulations were run, where different compounds were removed from the mechanism (benzene, toluene, xylenes, and the rest of aromatics).*

*In Figures 1 and 2, the sensitivity simulations are compared with the AROM scenario.*

*In general anthropogenic emissions dominate the impacts over biomass burning and biogenic sources. These emissions also have major relevance on the northern hemisphere for $NO_2$, $NO_3$, and Glyoxal. As expected, biomass burning sources have a large impact on specific areas: Siberia, central Africa, and the Amazonian forest. Increases of $NO_2$ above 10% are estimated in central Africa when biomass burning sources are missing, while Glyoxal decreases more than 10% in central Africa. Biogenic emissions are of minor relevance for the impacts on any species.*

*Among benzene, toluene, xylenes, and the rest of aromatics the first species has the lowest relevance due to its low reactivity (among the aforementioned aromatics). Toluene is the most important source for glyoxal overall and is also relevant for $NO_3$ in remote areas. Xylenes seem to have generally a small influence on the studied tracers. Interestingly, the rest of aromatics appear to lead the changes in OH, $NO_2$, $NO_3$ and glyoxal (this last only in continental areas) in areas dominated by biomass burning.*

**The study is conducted at a rather coarse horizontal model resolution (1.875 degrees). At this resolution urban chemistry cannot be represented, yet aromatics play an important role specifically in urban regimes. The authors themselves state in Line 260 that the coarse resolution limits the representation of urban chemistry. So why was the study not done at higher model resolution or with a higher resolution regional model?**

*It is well known that aromatics have strong impact on local scale (Tie et al., 2007; Stroud et al., 2008; Jaars et al., 2014), and therefore any high resolution urban model simulation would not have added anything to the already present knowledge. On the other side, the aim of this work is to find the chemical impact of aromatics at global scale, as this was never really quantified before for single aromatic species. The T63 resolution was chosen as the highest resolution possible with an affordable computation cost.*

**Has the aromatic chemistry in EMAC been compared to a fully explicit chemical scheme? If so, how well does it represent the full chemistry? The authors list a large range of uncertainties that impact the presented results and reduce confidence in the value of the conclusions.**

*The only chemical scheme that can be considered a "fully explicit scheme" for aromatics is the MCM mechanism (Bloss et al., 2005b). A model comparison between the original MCM and the mechanism used in this study has been done using the CAABA/MECCA box model and is fully described in Cabrera-Perez (2017). This information will be added to the methods section of the manuscript, summarizing the main results, i.e.:*

*"The aromatic mechanism has been evaluated against the MCM with the CAABA/MECCA box model in Cabrera-Perez (2017). The comparison shows that the aromatic mechanism consistently simulates ozone, OH and VOCs, with relative differences below 10% . The largest differences affect $NO_3$ and HONO, which are expected, due to the channels added in the aromatic mechanism including the phenoxy radical reactions and nitrophenol photolysis, respectively."*

**I agree that emissions in general can pose, in parts, large uncertainties. Have the authors looked at other inventories (HTAP, RETRO, NEIs, ...)? What was the reason for selecting IPCC emissions?**

*HTAP offers no speciation of organics, but rather a lumped NMVOCs emissions. We relied on the speciation provided by IPCC so to assure the correct regional emissions distribution (although this does not guarantee the global total emissions to be correct). Finally, as the IPCC and RETRO use the same speciation algorithm, no real differences are present between the two dataset, in this sense. Finally, RETRO covers only up to 2000, while we were interested in the year 2005 as the biomass burning database (GFAS) covers only from 2003 onwards. Finally IPCC emissions offers the advantages to have a long time coverage (1850-2100) when the historical data and the projections are used.*

**What is the reason for the high ozone bias in ECAM? Could the chemical scheme be a cause?**

*This issue in the EMAC model has been addressed in previous work Jöckel et al. (2016); Righi et al. (2015) this is the evaluation of MESSy2. It is not clear what is the cause of this bias, but our work clarifies that the bias is not due by the lack of details in the NMVOCs chemistry.*

**Line362: Do the authors imply that MCMV3.1 is incorrectly representing ozone chemistry or that the lumped chemical scheme that was developed based on MCMv3.1 and used in the current study has problems? If so, how much can one trust the derived sensitivity of tropospheric chemistry to aromatics? Does including HONO help with this problem? Has this been tested in a box model? Given the large role of SOA in aromatic chemistry, should such a study not have been conducted with a more detailed aerosol scheme that represents SOA formation.**

*We are confident on the results from MCMv3.1. In our work we summarize what Bloss et al. (2005a) pointed as limitations of MCM when simulating ozone mixing ratios. As mentioned in Cabrera-Perez et al. (2016), there were limitations in order to evaluate xylene concentrations, but the results show that in general EMAC is able to represent observations. The results of our study suggest that aromatics have small influence at the global scale, and therefore the limitations by MCM do not lead to large errors in the global scale. The HONO channels included in the model do not have a large effect on the sensitivity runs.*

**There is zero evaluation of the model with observations. Aromatic species themselves have been evaluated in Cabrera-Perez (2016), but what about e.g. O3 or CO?**

*There are several studies covering the evaluation of O3, CO and several organic species. (Jockel et al 2010, gmd, pozzer et al 2010). As mentioned before, the basic chemical mechanism is the one used in many model evaluation's works. Here detail comparison for O3 can be found in Jöckel et al. (2006); Jöckel et al. (2010); Jöckel et al. (2016), for CO in Yoon and Pozzer (2014) and for CO and other tracers (VOCs) in Pozzer et al. (2012b, 2010, 2012a). We do not believe that adding a detail evaluation of these tracers would give additional informations beside what is already present in the literature.*

*The following text will be added to the manuscript: "Besides the model evaluation for aromatics EMAC has been extensively evaluated for others species, including studies on O3 Jöckel et al. (2006); Jöckel et al. (2010); Jöckel et al. (2016), CO in Yoon and Pozzer (2014) and, CO and other tracers (VOCs) in Pozzer et al. (2007, 2010, 2012a)."*

**Minor Comments: Line 55 (and also Line 195): This is a very simplified description of ozone chemistry. High NOx regimes do not equate net ozone formation as at very high NOx concentrations ozone can actually be lost. The actual urban chemistry cannot be resolved at the coarse model resolution applied in here.**

*We fully agree that in this description effects of ozone titration are somehow neglected. Although the description given would be essential for the understanding of the manuscript, we understand that it should give a complete description, and therefore will be extended in the revised version. We introduced the following in the Introduction:*

*"When elevated ozone and NO concentrations are simultaneously present, another possible reaction occurs, the ozone titration (e.g. Kley et al., 1994):* $NO + O_3 \rightarrow NO_2 + O_2$*"*

**Line125: Given how closely this manuscript is linked to (Cabrera-Perez et al., 2016) I suggest referencing some of the major findings of this study as they apply to this work.**

*The following text has been added:*

*"Aromatic emissions are dominated by anthropogenic sources, followed by biomas burning emissions and finally biogenic emissions play a minor role. The largest sink of aromatics is chemical oxidation, being dry deposition a minor sink and wet deposition a negligible process. The EMAC model is able to represent the spatial distribution and annual cycle of background stations for benzene and toluene.Benzene and toluene have very low bias and root mean square error (below 50%) when compared to the observations, however higher discrepancies are present for toluene when representing the annual cycle. The complete description of the model setup including emissions, the chemical mechanism used, and the evaluation of the AROM scenario—are included in Cabrera-Perez et al. (2016). The complete set of chemical reactions can be found in the supplementary information (supplement in Cabrera-Perez et al., 2016).. The aromatic mechanism has been evaluated against the MCM with the CAABA/MECCA box model in Cabrera-Perez (2017). The comparison shows that the aromatic mechanism consistently simulates ozone, OH and VOCs, with relative differences below 10% . The largest differences*

115 *affect* $NO_3$ *and* $HONO$*, which are expected, due to the channels added in the aromatic mechanism including the phenoxy radical reactions and nitrophenol photolysis, respectively. Besides the model evaluation for aromatics EMAC has been extensively evaluated for others species, including studies on O3 Jöckel et al. (2006); Jöckel et al. (2010); Jöckel et al. (2016), CO in Yoon and Pozzer (2014) and, CO and other tracers (VOCs) in Pozzer et al. (2012b, 2010, 2012a).*"

120 **Figure 5 and Figure 6: Is this daytime? Daytime and Nighttime? Most Figure Captions require more details on what is shown.**

*We thanks the referee for pointing this out. More details have been added to the labels.*

| | |
|---|---|
| Caption in Fig 5 | Daytime OH relative difference (expressed in percent) in the boundary layer between the *AROM* and *REF* scenarios. In black, values for the northern hemisphere. In red, values for the southern hemisphere. |
| Caption in Fig 6. | Daytime $O_3$ relative difference (expressed in percent) in the boundary layer between the *AROM* and *REF* scenarios. In black, values for the northern hemisphere. In red, values for the southern hemisphere. |
| Caption in Fig 7 | Surface daytime $NO_2$ (left) and $NO_3$ (right) relative difference between aromatic and no-aromatic scenarios expressed in %. |
| Caption in Fig 8 | Surface daytime relative difference ((*No phenoxy channel-AROM*) / *AROM*) expressed as a percent for ozone (left) and $NO_3$ (right). |
| Caption in Fig 9 | Top. Annual average daytime surface mixing ratios of HCHO in $\mathrm{ppt}$. Bottom, daytime surface relative difference in %. |
| Caption in Fig 10 | Top. Annual average surface daytime mixing ratios of glyoxal in $\mathrm{ppt}$. Bottom, daytime surface relative difference in %. |

**The authors discuss the role of a number of intermediate species. I suggest adding the chemical reactions for the aromatics as included in MESSy in the Supplement or at least include a reference to where the**
125 **chemical reactions can be looked up.**

*A reference to the supplement of (Cabrera-Perez et al., 2016), has been added in the methods sections:.*

*"The complete set of chemical reactions can be found in the supplementary information (supplement in Cabrera-Perez et al., 2016)."*

**References**

130    Bloss, C., Wagner, V., Bonzanini, A., Jenkin, M. E., Wirtz, K., Martin-Reviejo, M., and Pilling, M. J.: Evaluation of detailed aromatic mechanisms (MCMv3 and MCMv3.1) against environmental chamber data, Atmospheric Chemistry and Physics, 5, 623–639, doi:10.5194/acp-5-623-2005, http://www.atmos-chem-phys.net/5/623/2005/, 2005a.

Bloss, C., Wagner, V., Jenkin, M. E., Volkamer, R., Bloss, W. J., Lee, J. D., Heard, D. E., Wirtz, K., Martin-Reviejo, M., Rea, G., Wenger, J. C., and Pilling, M. J.: Development of a detailed chemical mechanism (MCMv3.1) for the atmospheric

135    oxidation of aromatic hydrocarbons, Atmospheric Chemistry and Physics, 5, 641–664, doi:10.5194/acp-5-641-2005, https://www.atmos-chem-phys.net/5/641/2005/, 2005b.

Cabrera-Perez, C. D.: Simple Monocyclic Aromatic Compounds from a Global Scale Perspective, Ph.D. thesis, Johannes Gutenberg-Universität Mainz, 2017.

Cabrera-Perez, D., Taraborrelli, D., Sander, R., and Pozzer, A.: Global atmospheric budget of simple monocyclic aro-

140    matic compounds, Atmospheric Chemistry and Physics Discussions, 2016, 1–25, doi:10.5194/acp-2015-996, http://www.atmos-chem-phys-discuss.net/acp-2015-996/, 2016.

Jaars, K., Beukes, J., van Zyl, P., Venter, A., Josipovic, M., Pienaar, J., Vakkari, V., Aaltonen, H., Laakso, H., Kulmala, M., et al.: Ambient aromatic hydrocarbon measurements at Welgegund, South Africa, Atmospheric Chemistry and Physics, 14, 7075–7089, 2014.

145    Jöckel, P., Tost, H., Pozzer, A., Brühl, C., Buchholz, J., Ganzeveld, L., Hoor, P., Kerkweg, A., Lawrence, M. G., Sander, R., Steil, B., Stiller, G., Tanarhte, M., Taraborrelli, D., van Aardenne, J., and Lelieveld, J.: The atmospheric chemistry general circulation model ECHAM5/MESSy1: consistent simulation of ozone from the surface to the mesosphere, Atmospheric Chemistry and Physics, 6, 5067–5104, doi:10.5194/acp-6-5067-2006, https://www.atmos-chem-phys.net/6/5067/2006/, 2006.

Jöckel, P., Kerkweg, A., Pozzer, A., Sander, R., Tost, H., Riede, H., Baumgaertner, A., Gromov, S., and Kern, B.: Development

150    cycle 2 of the modular earth submodel system (MESSy2), Geoscientific Model Development, 3, 717–752, 2010.

Jöckel, P., Tost, H., Pozzer, A., Kunze, M., Kirner, O., Brenninkmeijer, C. A. M., Brinkop, S., Cai, D. S., Dyroff, C., Eckstein, J., Frank, F., Garny, H., Gottschaldt, K.-D., Graf, P., Grewe, V., Kerkweg, A., Kern, B., Matthes, S., Mertens, M., Meul, S., Neu-maier, M., Nützel, M., Oberländer-Hayn, S., Ruhnke, R., Runde, T., Sander, R., Scharffe, D., and Zahn, A.: Earth System Chemistry integrated Modelling (ESCiMo) with the Modular Earth Submodel System (MESSy) version 2.51, Geoscien-

155    tific Model Development, 9, 1153–1200, doi:10.5194/gmd-9-1153-2016, https://www.geosci-model-dev.net/9/1153/2016/, 2016.

Kley, D., Geiss, H., and Mohnen, V. A.: Tropospheric ozone at elevated sites and precursor emissions in the United States and Europe, Atmospheric Environment, 28, 149 – 158, doi:https://doi.org/10.1016/1352-2310(94)90030-2, http://www.sciencedirect.com/science/article/pii/1352231094900302, 1994.

160    Pozzer, A., Jöckel, P., Tost, H., Sander, R., Ganzeveld, L., Kerkweg, A., and Lelieveld, J.: Simulating organic species with the global atmospheric chemistry general circulation model ECHAM5/MESSy1: a comparison of model results with observations, Atmospheric Chemistry and Physics, 7, 2527–2550, 2007.

Pozzer, A., Pollmann, J., Taraborrelli, D., Joeckel, P., Helmig, D., Tans, P., Hueber, J., and Lelieveld, J.: Observed and simulated global distribution and budget of atmospheric C$_2$-C$_5$ alkanes, Atmospheric Chemistry and Physics, 10, 4403–4422, 2010.

165    Pozzer, A., Meij, A. d., Pringle, K., Tost, H., Doering, U., Aardenne, J. v., and Lelieveld, J.: Distributions and regional budgets of aerosols and their precursors simulated with the EMAC chemistry-climate model, Atmospheric Chemistry and Physics, 12, 961–987, 2012a.

Pozzer, A., Zimmermann, P., Doering, U. M., van Aardenne, J., Tost, H., Dentener, F., Janssens-Maenhout, G., and Lelieveld, J.: Effects of business-as-usual anthropogenic emissions on air quality, Atmospheric Chemistry and Physics, 12, 6915–6937,

170    doi:10.5194/acp-12-6915-2012, https://www.atmos-chem-phys.net/12/6915/2012/, 2012b.

Righi, M., Eyring, V., Gottschaldt, K.-D., Klinger, C., Frank, F., Jöckel, P., and Cionni, I.: Quantitative evaluation of ozone and selected climate parameters in a set of EMAC simulations, Geoscientific Model Development, 8, 733–768, doi:10.5194/gmd-8-733-2015, https://www.geosci-model-dev.net/8/733/2015/, 2015.

Stroud, C., Morneau, G., Makar, P., Moran, M., Gong, W., Pabla, B., Zhang, J., Bouchet, V., Fox, D., Venkatesh, S.,

175    Wang, D., and Dann, T.: OH-reactivity of volatile organic compounds at urban and rural sites across Canada: Evaluation of air quality model predictions using speciated VOC measurements, Atmospheric Environment, 42, 7746 – 7756, doi:https://doi.org/10.1016/j.atmosenv.2008.05.054, http://www.sciencedirect.com/science/article/pii/S1352231008005104, 2008.

Tie, X., Madronich, S., Li, G., Ying, Z., Zhang, R., Garcia, A. R., Lee-Taylor, J., and Liu, Y.: Characterizations of

180    chemical oxidants in Mexico City: A regional chemical dynamical model (WRF-Chem) study, Atmospheric Environment, 41, 1989 – 2008, doi:https://doi.org/10.1016/j.atmosenv.2006.10.053, http://www.sciencedirect.com/science/article/pii/S1352231006010399, 2007.

Yoon, J. and Pozzer, A.: Model-simulated trend of surface carbon monoxide for the 2001–2010 decade, Atmospheric Chemistry and Physics, 14, 10 465–10 482, doi:10.5194/acp-14-10465-2014, https://www.atmos-chem-phys.net/14/10465/2014/,

185    2014.

[Figure]

**Figure 1.** Surface annual mean relative differences between aromatic and the different scenarios expressed in %.

[Figure]

**Figure 2.** Surface annual mean relative differences between aromatic and the different scenarios expressed in %.

---

## Author Comment (AC2) · 2 May 2018

The comment was uploaded in the form of a supplement:
https://www.atmos-chem-phys-discuss.net/acp-2017-928/acp-2017-928-AC2-supplement.pdf

---

## Author Comment (AC3) · 2 May 2018

The comment was uploaded in the form of a supplement:
https://www.atmos-chem-phys-discuss.net/acp-2017-928/acp-2017-928-AC3-supplement.pdf